



# Direct measurements of NO₃-reactivity in and above the boundary layer of a mountain-top site: Identification of reactive trace gases and comparison with OH-reactivity.

Jonathan M. Liebmann[1], Jennifer B. A. Muller[2], Dagmar Kubistin[2], Anja Claude[2], Robert Holla[2], Christian Plaß-Dülmer[2], Jos Lelieveld[1] and John N. Crowley[1]

[1]Atmospheric Chemistry Department, Max Planck Institut für Chemie, 55128, Mainz, Germany
[2]Meteorologisches Observatorium Hohenpeissenberg, Deutscher Wetterdienst, 82383, Hohenpeissenberg, Germany

*Correspondence to:* John Crowley (john.crowley@mpic.de)

**Abstract.** We present direct measurements of the summertime, total reactivity of NO₃ towards organic trace gases, $k_{\mathrm{OTG}}^{\mathrm{NO_3}}$, at

a rural mountain site (988 m a.s.l.) in southern Germany in 2017. The diel cycle of $k_{\mathrm{OTG}}^{\mathrm{NO_3}}$ was strongly influenced by local

meteorology with reactivity high during the day (values of up to 0.3 s⁻¹) but usually close to the detection limit (0.005 s⁻¹) at

night when the measurement site was in the residual layer / free troposphere. Daytime values of $k_{\mathrm{OTG}}^{\mathrm{NO_3}}$ were sufficiently large

that the loss of NO₃ due to reaction with organic trace gases competed with its photolysis and reaction with NO. Within

experimental uncertainty, monoterpenes and isoprene accounted for all of the measured NO₃-reactivity. Averaged over the

daylight hours, more than 25% of NO₃ was removed via reaction with biogenic volatile organic compounds (BVOCs),

implying a significant daytime loss of NOₓ and formation of organic nitrates due to NO₃ chemistry. Ambient NO₃

concentrations were measured on one night and were comparable to those derived from a stationary state calculation using

measured values of $k_{\mathrm{OTG}}^{\mathrm{NO_3}}$. We present and compare the first simultaneous, direct-reactivity measurements for the NO₃ and

OH radicals. The decoupling of the measurement site from ground level emissions resulted in lower reactivity at night for

both radicals, though the correlation between OH- and NO₃-reactivity was weak as would be anticipated given their

divergent trends in rate constants with many organic trace gases.



## 1 Introduction

Hydroxyl (OH) and nitrate radicals (NO$_3$) play a centrally important role in cleansing the atmosphere of trace gas emissions resulting from both anthropogenic and biogenic activity (Lelieveld et al., 2004; Lelieveld et al., 2016; Ng et al., 2017). Whereas OH is largely photochemically generated and present at its highest concentrations during the day, NO$_3$ is generated

through the oxidation of NO$_2$ by O$_3$ and, due to its rapid photolysis and reaction with NO, is present mainly at night. A further important difference in the roles of OH and NO$_3$ in the atmosphere is related to the mechanism of their reactions. NO$_3$ reacts rapidly via electrophilic addition to unsaturated organic trace gases but reacts comparatively slowly (via H-abstraction) with saturated organics. In the presence of O$_2$, the initial addition step results in the formation of nitrooxyalkyl peroxy radicals, which can react with HO$_2$, NO, NO$_2$ or NO$_3$ to form multifunctional peroxides and organic nitrates (Fry et

al., 2014; Ng et al., 2017).

OH can react both by addition and H-abstraction to organic and inorganic trace gases and may be considered to be more reactive and much less selective than the NO$_3$ radical. The distinct reaction modes leads to significant differences in the lifetimes of both radicals, which for OH are typically less than 1s and for NO$_3$ can exceed 1 hour (Wayne et al., 1991; Atkinson, 2000; Atkinson and Arey, 2003a; Brown and Stutz, 2012; Liebmann et al., 2018). Maximum daytime

concentrations of OH are typically less than 1 pptv, whereas NO$_3$ has been observed at the 10s to 100s of pptv levels during nighttime (Noxon et al., 1978; Sobanski et al., 2016; Ng et al., 2017).

The large NO$_3$ mixing ratios at nighttime and the large rate constants for reaction of NO$_3$ with several unsaturated, biogenic VOCs result in NO$_3$ being the dominant sink of many BVOCs (Wayne et al., 1991; Atkinson, 2000; Atkinson and Arey, 2003a, b; Long et al., 2011; Brown and Stutz, 2012; Liebmann et al., 2017) especially those whose emission is mainly

temperature dependent and continues at nighttime, e.g. monoterpenes (Hakola et al., 2012). The importance of NO$_3$ on a global scale is highlighted by the fact that forest ecosystems (covering around 9% of the world's surface) annually release ≈ 1000 Tg of biogenic volatile organic compounds (BVOC, e.g. isoprene (2-methyl-1,3-butadiene), monoterpenes (C$_{10}$H$_{16}$) and sesquiterpenes (C$_{15}$H$_{24}$)) into the Earth's atmosphere (Guenther et al., 2012; Bastin et al., 2017). BVOCs have a strong impact on the atmospheric radical budget, the NO$_x$ cycle (Hakola et al., 2003; Holzke et al., 2006; Nölscher et al., 2013) as

well as on the formation and growth of organic particles (Jaoui et al., 2013; Lee et al., 2016; Ng et al., 2017) hence understanding their lifetime and fate is essential for predicting atmospheric processes and climate change (Lelieveld et al., 2008; Lelieveld et al., 2016). In addition, NO$_3$ is an intermediate in the step-wise oxidation of NO to N$_2$O$_5$ (R1-R2, R4) and its lifetime with respect to reaction with biogenic trace gases (R6) impacts on NO$_x$ levels and thus on photochemical O$_3$ formation from NO$_2$ photolysis.

$$NO + O_3 \rightarrow NO_2 + O_2 \tag{R1}$$

$$NO_2 + O_3 \rightarrow NO_3 + O_2 \tag{R2}$$

$$NO + NO_3 \rightarrow 2NO_2 \tag{R3}$$

$$NO_3 + NO_2 + M \rightarrow N_2O_5 + M \tag{R4}$$



| | | |
|---|---|---|
| $N_2O_5 + M$ | $\rightarrow NO_3 + NO_2 + M$ | (R5) |
| $NO_3 + BVOC\ (O_2)$ | $\rightarrow \rightarrow$ organic nitrates (gas, particle) | (R6) |
| $N_2O_5$ + particle | $\rightarrow 2\ NO_3^-$ or $(NO_3^- + ClNO_2)$ | (R7) |

The organic nitrates formed in the multi-step reaction (R6) can transfer to the particle phase or be lost through deposition;

$N_2O_5$ formed in (R4) can react with aqueous particles to form particulate nitrate and/or $ClNO_2$ (R7) (Osthoff et al., 2008; Phillips et al., 2012; Bannan et al., 2015; Phillips et al., 2016) thus reducing the rate of photochemical $O_3$ production (Dentener and Crutzen, 1993). The absolute and relative fluxes through (R6) and (R7) thus control to some extent the lifetime of $NO_x$.

Direct $NO_3$-reactivity measurements have recently become possible (Liebmann et al., 2017) and the first deployment in a

forested region revealed a large $NO_3$-reactivity at canopy height, not all of which could be accounted for by simultaneous measurements of a large suite of organic trace gases (Long et al., 2011) pointing towards unmeasured monoterpenes as well as sesquiterpenes likely to be responsible. The difference between the observed (or derived) $NO_3$-reactivity and that calculated from summing loss rates for a set of reactive trace gases is generally termed "missing reactivity" as frequently reported for OH (Nölscher et al., 2012). Previous work on $NO_3$-reactivity has also revealed a strong meteorological influence

on the $NO_3$ lifetime, especially when air masses are decoupled from the surface layer in which reactive trace gases (NO and BVOC) are emitted at night (Brown et al., 2007b; Brown et al., 2011; Long et al., 2011; Sobanski et al., 2016).

In this paper we describe direct measurements of the $NO_3$-reactivity in ambient air on a rural mountain site in southern Germany and interpret the data based on measured VOCs and in terms of the underlying meteorological situation. We also compare $NO_3$-reactivity to simultaneous measurements of OH-reactivity over the same period.

**2 Site description and methods**

During the period 20.07.17 to 6.08.17 $NO_3$-reactivity measurements were conducted in parallel with ongoing observations at the Meteorological Observatory Hohenpeissenberg (MOHp) in Bavaria, southern Germany. The observatory is a meteorological monitoring and Global Atmosphere Watch site, operated by the German Meteorological Service (DWD). It is located on the Hohenpeissenberg mountain (988 m a.s.l.), 300-400 m above the surrounding countryside about 40 km from

the northern rim of the Alps and has been the location of several intensive field campaigns (Plass-Dulmer et al., 2002; Birmili et al., 2003; Handisides et al., 2003; Mannschreck et al., 2004; Bartenbach et al., 2007; Hock et al., 2008; Novelli et al., 2017). The vegetation around the measurement site consists of coniferous trees and beeches growing on the slopes of the mountain while grassland and marshes are dominant in the valley. Tourism related vehicular emissions represent a potential source of local anthropogenic pollution especially at the weekends. The nearest city, Munich, is about 70 km to the

northeast.

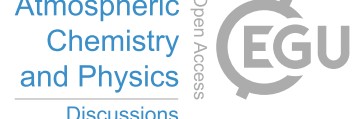

Trace gases were sampled into the NO$_3$-reactivity and NO$_2$-CRD instruments through 1-inch outer-diameter PFA tubing (20 m long, operated at a flow of 40 dm$^3$ min$^{-1}$) located 1.5m above the roof, directly next to the VOC inlet. The inlet was circa 3 m distance from the inlet used for the other NO$_x$ measurements and circa 2 m distance from the OH-reactivity inlet.

## 2.1 NO$_3$-reactivity measurements

The NO$_3$-reactivity instrument was operated in a laboratory located in the 3rd floor of the of the MOhp station building at the Hohenpeissenberg. Air samples were drawn at a flow rate of 2900 cm$^3$ (STD) min$^{-1}$ through a 2 µm membrane filter (Pall Teflon) and 4 m of PFA tubing (6.35 mm OD) from the centre of the bypass flow (see above) resulting in 7.5 s residence time for the transport of air from the sampling point. During night time (≈19:00-03:50 UTC) ambient air samples were drawn through a heated glass flask (35 °C, residence time 20 s) to destroy ambient N$_2$O$_5$ and NO$_3$ which would potentially

interfere with the reactivity measurements. Operational details of the instrument were recently described by Liebmann et al. (2017). NO$_3$ radicals were generated by mixing NO$_2$ and O$_3$ at elevated pressure (1.5 bar, ≈ 5 minutes reaction time) and passing the mixture through an oven at ≈ 100 °C to convert all N$_2$O$_5$ to NO$_3$ (R2-R5). The effluent from the oven was mixed with either zero-air or ambient air in a flow-tube thermostatted to 21 °C to yield a typical (initial) NO$_3$ mixing ratios of 40-60 pptv.

After a fixed reaction time, the remaining NO$_3$ was detected by cavity-ring-down spectroscopy (CRDS) at 662 nm. The lower pressure at the top of the Hohenpeissenberg station (903 ± 8 hPa) meant that the reaction time was reduced from 10.5 s as previously reported (Liebmann et al., 2017), to 9.5 s. The measurement cycle was typically 400 s for synthetic air and 1200 s for ambient air, with intermittent signal zeroing (every ≈ 100 s) by addition of NO. The fractional loss of NO$_3$ in ambient air compared to zero air was converted to a reactivity via numerical simulation of a simple reaction scheme

(Liebmann et al., 2017) using measured amounts of NO, NO$_2$ and O$_3$. The parameter obtained, $k_{OTG}^{NO_3}$, is a NO$_3$ loss rate constant from which contributions by NO and NO$_2$ have been removed, and thus refers to reactive loss to organic trace gases (OTG) only. Throughout the manuscript, NO$_3$-reactivity and $k_{OTG}^{NO_3}$ are equivalent terms, with units of s$^{-1}$. The upper measurement limit to $k_{OTG}^{NO_3}$ was 45 s$^{-1}$, achieved by automated, dynamic dilution of the air sample. The lower limit was 0.005 s$^{-1}$, defined by the stability of the NO$_3$ source. Online calibration of the reactivity using an NO standard was performed every

≈2 hours for 10 min. The uncertainty of the measurement was between 0.015 and 0.205 s$^{-1}$, depending mainly on dilution accuracy, NO levels and stability of the NO$_3$ source as described by Liebmann et al. (2017). Since its first description in Liebmann et al. (2017), the instrument has been extended with a further cavity to measure mixing ratios of NO$_2$ (see below).

## 2.2 NO$_2$, NO and O$_3$ measurements

Since its first deployment, the NO$_3$-reactivity instrument described by Liebmann et al. (2017, 2018) has been extended with a further cavity to measure NO$_2$. This is described here for the first time and thus in detail. The CRDS measurement of NO$_2$





uses a 2500 Hz, square-wave modulated, 40 mW laser-diode located in a Thor Labs LDM 21 housing and thermally stabilized at 36 °C using a Thor Labs ITC 510 Laser-Diode Combi Controller to produce light at 405 nm (0.5 nm full-width at half-maximum). The laser-diode emission is first directed through an optical isolator (Thorlabs IO-3D-405-PBS), focused by a lens (Thorlabs C340TMD-A) into the optical-fibre (0.22 NA, 50 µm core, 400-2400 nm) and then collimated (Thor

Labs FiberPort Collimator PAF-X-7-A) to a beam diameter of about 6 mm before entering the cavity. Part of the laser emission was directed to an Ocean Optics spectrograph to continuously measure the laser emission spectrum.

The $NO_2$ cavity (Teflon-coated glass (DuPont, FEP, TE 9568), length 70 cm, volume 79 cm$^3$) was operated at 30 °C at a flow rate of 3000 cm$^3$ (STP) min$^{-1}$ resulting in a residence time of approximately 1 s. To remove particles, air was drawn through a 2 µm membrane filter (Pall Teflon) from the centre of the same high-flow bypass used for the $NO_3$-reactivity

measurements. Light exiting the cavities through the rear mirror was detected by a photomultiplier (Hamamatsu E717-500) which was screened by a 405 nm interference filter. The pre-amplified PMT signal was digitized and averaged with a 10 MHz, 12 bit USB scope (Picoscope 3424) which was triggered at the laser modulation frequency of 2500 Hz.

The ring-down constant in the absence of $NO_2$ was obtained by adding zero air every 30 points of measurement for approximately 15 s. The $L/d$ ratio (the ratio of the distance between the cavity mirrors, $L$, and the length of the cavity that is

filled by absorber, $d$) was determined as described previously (Schuster et al., 2009; Crowley et al., 2010) and was 1.00 ± 0.03. Inverse decay-constants in dry zero-air at 660 Torr were usually between 28 and 31 µs indicating optical path lengths of ≈ 8-9 km. The measurement precision (6 s integration) was circa 150 pptv. The cavity was not pressure stabilized leading to a pressure difference of circa 2.5 Torr when switching from ambient air to zero measurements. The data was corrected for the change in Rayleigh scattering resulting from the pressure difference (typically 120 pptv) and also different relative

humidities (typically 60 to 100 pptv) when switching from ambient to zero-air measurement as described by Thieser et al. (2016). The laser spectrum was measured every hour and used to calculate an effective cross-section (≈ 6.00 x 10$^{-19}$ cm$^2$ molecule$^{-1}$) using a literature absorption spectrum (Voigt et al., 2002). The overall uncertainty of the measurement is mainly determined by the uncertainty in the cross section (6%). Other contributions are from $NO_2$ formation (from reaction of NO with $O_3$ in the inlet lines, ≈ 0.5%) and the correction for humidity and pressure differences (5%), and an error in the L/$d$ ratio

(2%), giving an estimated uncertainty of 9%. The detection limit of the instrument can be estimated from the variability in the zeros and was usually around 150 pptv.

$NO_2$ measurements were made from the 20$^{th}$ of July to the 4$^{th}$ of August with breaks from the 27$^{th}$ of July to the 2$^{nd}$ August and from the 4$^{th}$ to the 6$^{th}$ due to instrumental problems. $NO_2$ mixing ratios were corrected for its formation (R1) during transport from the roof top inlet to the cavity (≈ 7.5 s).

Two commercially available instruments operated permanently at the site also provided measurements of $NO_2$ and NO. These were a cavity-phase-shift (CAPS) instrument for $NO_2$ measurement and a chemiluminescence device (CLD) for $NO_2$ and NO. The CAPS (Aerodyne, Ambient Monitor Version 2012) had a detection limit of 270 pptv (3 σ in 1 min integration time) and an uncertainty of 10% (1 min integration time). The CLD (ECO PHYSICS, Model AL 770 pptv) uses chemiluminescence from the reaction of NO with ozone in combination with a blue-light converter to convert $NO_2$ to NO.




The instrument was routinely calibrated once a week (10 ppmv ± 5% NO in $N_2$, Riessner, Germany. Deviations between two calibrations are typically well below 3%. Detection limits during the intensive were 11 pptv for NO, 16 pptv for $NO_2$ (3 σ in 1min integration time) and median uncertainties are 27 pptv (7%) for NO and 70 pptv (10%) for $NO_2$ (2 σ at 1 ppbv in 1min integration time). Corrections were applied to take into account NO loss and $NO_2$ formation due to further reactions

involving ozone in the inlet tubing.

A comparison of the three $NO_2$ measurements instruments is given in Fig. S1 of the supplementary information which plots the $NO_2$ mixing ratios (averaged over 60 s) of the CLD and CAPS instruments versus the CRDS. A least-squares fit (considering errors in both parameters) to the plot of $NO_2$ (CLD) versus $NO_2$ (CRDS) has a slope of 0.94 ± 0.25 and an

intercept of 0.00 ± 0.04. For the plot of $NO_2$ (CAPS) versus $NO_2$ (CRDS) comparison we derive a slope of 0.95 ± 0.02 and an intercept of 0.00 ± 0.02. Within combined uncertainty, the $NO_2$ measurements are thus in agreement. The $NO_2$ mixing ratios used as input to calculate the $NO_3$-reactivity were taken from the CRDS instrument, with data gaps filled by CAPS measurements.

Ozone was monitored with a UV absorption instrument (Thermo Environmental Instruments Inc., Model TECO 49C) which

is calibrated at regular intervals with a transfer standard (TECO 49 PS). The uncertainty in the ozone mixing ratio is 1.2 ppbv or 2% (2 σ in 1 hour).

### 2.3 $NO_3$ measurements

For the measurement of ambient $NO_3$, a 10 m length of PFA-tubing (3/8-inch outer diameter) was installed on the top of the

building, circa 10 cm from the VOC inlet. A 2 μm pore PTFE filter (47 mm in diameter, replaced every hour) in a PFA filter holder was located at the end of the inlet. The tubing was connected to a bypass pump operated at 20 $dm^3$ (STD) $min^{-1}$ to reduce the residence time. The sample flow through the cavity was increased to 8 $dm^3$ (STD) $min^{-1}$ to reduce the $NO_3$ residence time within the cavity. $NO_3$ mixing ratios were recorded every 6 s (3600 ring-downs co-added) with zeroing by titration (no addition) every 15 data points. The $NO_3$ transmission through the inlet (67 ± 15%), filter and filter-holder (84 ±

10%) and cavity (88 ± 10%) were established post-campaign and used to correct the data. The overall uncertainty in the $NO_3$ measurements, including uncertainty in the absorption cross-section, was circa 35%.

### 2.4 OH-reactivity measurements

OH-reactivity measurements were conducted using a chemical ionization mass spectrometer (CIMS) in which OH radicals (generated by photolysis of $H_2O$ at 184.95 nm) are converted to $H_2SO_4$ (Berresheim et al., 2000; Schlosser et al., 2009). For

the derivation of OH-reactivity ($k_{total}^{OH}$), relative OH radical concentrations are measured at two fixed reaction times and a decay constant is derived assuming exponential behaviour. After correction for wall losses, as well as NO-induced HO$x$



recycling in the sample tube, ambient reactivities between 1 to 40 s$^{-1}$ are measureable. OH-reactivity measurements were made every 20 min throughout the measurement period. Measurements were discontinued during periods of precipitation and when the pinhole to the mass spectrometer vacuum system was blocked by insects. The instrument performs best if NO mixing ratios are below 4 ppbv and reactivities do not exceed 15 s$^{-1}$; for the measurements reported here, the mean uncertainty in the OH-reactivity was ± 1.2 s$^{-1}$ (or 46%). OH-reactivity calibration was carried out before and after the measurement period, and the calibration factor was applied to the whole dataset. The determination of the OH wall loss rate from zero reactivity measurement (null measurement) using synthetic air cylinders was not reliable and therefore the zero was estimated using nighttime measurements when sampling from above the boundary layer. Details are given in the supplementary information.

## 2.5 VOC measurements

A gas-chromatograph (GC-MS/FID model AGILENT 6890 with 5975 B inert XL MSD), was used for the detection of $C_5$–$C_{13}$ NMHCs and BVOC (Hoerger et al., 2015). In a custom-made pre-concentration unit, air was sampled at 30 °C on a 3-bed adsorption trap and, after a cryo-focussing step, injected onto the GC column (50m BPX-5). Subsequently, signals were detected with a mass spectrometer (MS) running in parallel with a flame ionization detector (FID). The instrument measured i.a. isoprene and a wide variety of monoterpenes with uncertainties ($2\sigma$) from 6 to 100% depending on the compound.

For the detection of light NMHCs ($C_2$–$C_8$), a GC-FID system (GC-1, Varian 3600 CX, FID detector) described in detail by Plass-Dülmer et al. (2002) was used. In both systems, an ozone scrubber (impregnated filter with $Na_2SO_3$) was used and water was removed from the sample air either by hydrophobic adsorbents ($C_5$–$C_{13}$) or a cold trap ($C_2$–$C_8$) prior to the pre-concentration step.

VOCs were sampled every hour for 15 min ($C_2$–$C_8$) respectively 20 min ($C_5$–$C_{13}$) by both instruments. During the rainy period from the 24$^{th}$ of July 12:00 UTC to the 27$^{th}$ of July 12:00 UTC, VOCs were only measured twice daily.

## 3 Results and discussion

Figure 1 displays the time series of NO$_3$-reactivity ($k_{OTG}^{NO_3}$) along with related trace gases and meteorological data obtained during the intensive. Sunrise was around 3:50 UTC and sunset at ≈19:00 UTC. Two mild days (T$_{max}$ = 20-25 °C) at the beginning of the campaign were followed by a 3 day period with heavy rains and maximum temperatures of around 10 °C followed by a warm period with temperatures up to 30 °C and occasional thunderstorms. The predominant wind direction was west-south-west with only minor contributions from other directions (Fig. S2). Wind speeds were generally around 2.5 to 7.5 ms$^{-1}$, increasing up to 15 ms$^{-1}$ during the rainy periods. The highest values of $k_{OTG}^{NO_3}$ were detected with north-easterly winds (Fig. S2) coincident with the warmest days of the campaign and the highest biogenic emissions (see section 4.2). Ozone mixing ratios were strongly correlated with temperature and ranged from 85 ppbv at the 1$^{th}$ of August during the warm, photochemically intense period to less than 20 ppbv during the cool, rainy period between 23$^{rd}$ and 28$^{th}$ of August.



$NO_x$ levels ($NO_x$ = NO + $NO_2$) during the intensive were generally between about 0.5 and 4 ppbv. The mixing ratios of NO, a trace gas which can potentially impact $NO_3$ lifetimes, were generally below the detection limit ($\approx$ 12 pptv) during most of the nights, increasing to maximum values of < 1 ppbv during the day. Occasional maxima of more than 1 ppbv NO were observed due to local traffic.

**3.1 $NO_3$ reactivity**

$NO_3$-reactivity was measured continuously during a three week intensive (20.07.2017 to 06.08.2017) with the exception of one night ($2^{nd}$ - $3^{rd}$ August) when, using the same instrument, $NO_3$ mixing ratios were measured instead. The full data set of $k_{OTG}^{NO_3}$ is reproduced in Fig. S3 of the supplementary information together with the corresponding 95% uncertainty limits, which take into account drifts in the zero signal, the stability of the $NO_3$ source, uncertainty in the dilution factors,
uncertainty of the NO and $NO_2$ mixing ratios as well as the corresponding rate constants.

As described above, during daytime the short $NO_3$ lifetime normally results in levels that are under the detection limit of most instruments, precluding estimation of the $NO_3$-reactivity via stationary-state calculations based on its mixing ratio and production rate. In contrast, our direct measurement enables us to derive the $NO_3$-reactivity over the full diel cycle. During the intensive, the 10-minute averaged values of $k_{OTG}^{NO_3}$ obtained ranged from below the detection limit (< 0.005 $s^{-1}$) to values
as high as 0.3 $s^{-1}$. Campaign averaged values were low ($\approx$ 0.01 $s^{-1}$) during nighttime but a factor of ten larger $\approx$ 0.1 $s^{-1}$ at 14:00 UTC (local 16:00) and more variable during daytime.

This observation is in stark contrast to the high relative night-time/daytime $NO_3$-reactivities we observed in a Boreal forest (Liebmann et al., 2018) and is related to very different meteorological conditions at the two sites. In the boreal forest, the canopy-level $NO_3$-reactivity was controlled by the rate of emission of biogenic VOCs into a nocturnal boundary layer of
varying height and stability. The elevated location of the Hohenpeissenberg observatory, located at a mountain-top above the surrounding countryside favoured sampling from the residual layer/free troposphere at nighttime. In the absence of turbulent exchange, the residual layer/free troposphere may become disconnected from the planetary boundary layer (PBL) and thus from ground-level emissions of reactive trace gases and may thus contain low levels of biogenic trace gases as well as low(er) levels of $NO_2$ and higher levels of ozone (Aliwell and Jones, 1998; Allan et al., 2002; von Friedeburg et al., 2002;
Stutz et al., 2004; Brown et al., 2007a; Brown et al., 2007b; Brown and Stutz, 2012). $NO_3$ lifetimes as long as 1 hour (using stationary-state analyses) have been reported for mountain sites when sampling air from above the nocturnal boundary layer (Brown et al., 2016; Sobanski et al., 2016).

During the Hohenpeissenberg intensive two distinct air-mass types were encountered at night, whereby values of $k_{OTG}^{NO_3}$ were either at (or below) the detection limit or well above it (named type 1 and type 2, respectively). Figure 2 displays a time
series of $k_{OTG}^{NO_3}$ over a single night ($29^{th}$ -$30^{th}$ July) in which a switch from type 2 to type 1 was observed. From early evening until shortly before 12:00 UTC, $NO_3$-reactivity was variable with values between circa 0.02 and 0.03 $s^{-1}$. A sharp reduction in $k_{OTG}^{NO_3}$ was then observed with values close to the detection limit until sunrise (04:00 UTC). The reduction in $k_{OTG}^{NO_3}$ was





accompanied by a drop in relative humidity (from 70% to 60%) and an increase in $O_3$ (46 to 52 ppbv), both clear indicators of sampling from the residual layer. At the same time, the wind speed increased and the temperature became more variable, indicating that the site was close to the inversion level. At first sunlight, turbulent mixing resulted in gradual connection of the boundary layer and overlying residual layer, leading to an increase in $k_{OTG}^{NO_3}$. Upslope winds caused by heating of the

mountainside may also have enhanced transport of air masses with high reactivity to the measurement site. Median, diel profiles of $NO_3$-reactivity on type 1 (altogether five) and type 2 nights (altogether 10) are shown in Fig. 3. With the exception of the very low reactivity during type1 nights, type 1 and type 2 have similar diel shapes and similar maximum reactivities.

## 3.2 $NO_3$-reactivity calculated from VOC measurements

In this section we assess the contribution of various VOCs to the observed $NO_3$-reactivity. The most abundant BVOCs were isoprene, sabinene, α-pinene and ß-pinene with maximum mixing ratios during the warm period around the 1[st] August. The time series of BVOC mixing ratios are displayed in Fig. S4 of the supplementary information. $k_{OTG}^{NO_3}$ is a total loss rate constant for chemical reactions of $[NO_3]$ with all organic trace gases present and can be compared to the summed loss rate constant ($k_{VOC}^{NO_3}$) (also in units of s[-1]) obtained from the concentrations of individual VOCs in the same air mass, $[C_i]$, and the

rate coefficient ($k_i$) :

$$k_{VOC}^{NO_3} = \sum k_i^{NO_3}[C_i] \tag{1}$$

Where $[C_i]$ is the measured BVOC concentration and $k_i$ the corresponding rate constant. Individual values of $k_{VOC}^{NO_3}$, calculated using rate constants from the IUPAC evaluation (IUPAC, 2017) or elsewhere in the literature (Shorees et al., 1991), are plotted with interpolated 20 min averages of $k_{OTG}^{NO_3}$ as a time series in the upper panel of Fig. 4.

The data are also displayed as a pie chart in the lower panel of Fig. 4 in which the contribution of individual biogenic trace-gases to the $NO_3$-reactivity are listed. Of the terpenoids, α-pinene contributed most to the overall $NO_3$-reactivity ($\approx$ 16%) followed by sabinene ($\approx$ 12%) with other individual BVOCs contributing less than 10%. VOCs such as methanol, acetaldehyde, ethanol, acetone, methylethylketone, alkanes and aromatics were also measured but not included in calculations of $k_{VOC}^{NO_3}$ as their summed contribution reached max. $1.5 \times 10^{-4}$ s[-1] and was on average $5 \times 10^{-5}$ s[-1]. As $k_{VOC}^{NO_3}$ and

$k_{OTG}^{NO_3}$ show a similar dependence on wind direction (Fig. S2) and because only BVOCs were used for the derivation of $k_{VOC}^{NO_3}$, we conclude that the high $NO_3$ reactivities measured in air masses arriving from the east and northeast are mostly from trace-gases of biogenic origin.

Type1 nights were characterized by very low BVOC mixing ratios, sometimes below the detection limit, whereas isoprene was still present. This observation is consistent with a long lifetime for isoprene in the residual layer at night (Brown et al.,

2007a) as the OH concentration is too low and the $NO_3$ reaction too slow to remove it efficiently. Under conditions of very low $NO_3$-reactivity, the fractional contribution of isoprene to the overall reactivity could increase to $\approx$ 100% (from typically





20% during the day). During type two nights (those with non-zero $NO_3$-reactivity) isoprene and monoterpenes were always detected and monoterpenes were the dominant reaction partners for $NO_3$.

The difference between $k_{OTG}^{NO_3}$ and $k_{VOC}^{NO_3}$ (i.e. the $NO_3$-reactivity not accounted for by measured VOCs) may be defined as "missing" reactivity ($s^{-1}$):

$$\text{missing } NO_3\text{-reactivity} = k_{OTG}^{NO_3} - k_{VOC}^{NO_3} \tag{2}$$

A plot of $k_{VOC}^{NO_3}$ versus $k_{OTG}^{NO_3}$ (see Fig. S5 of the supplementary information) has a slope of $1.55 \pm 0.15$ and an intercept of 0.005. This implies, on average, a missing reactivity of $\approx 34\%$ when $k_{OTG}^{NO_3} = 0.3 \text{ s}^{-1}$ and a missing reactivity of $\approx 50\%$ when $k_{OTG}^{NO_3} = 0.03 \text{ s}^{-1}$. However, both $k_{OTG}^{NO_3}$ and $k_{VOC}^{NO_3}$ are associated with some uncertainty, which needs to be rigorously assessed to test whether the missing reactivity is significant. To do this we propagated uncertainty in each of the terms $\sum k_i^{NO_3}[C_i]$ (mainly related to VOC measurements and assuming 15% uncertainty in the rate coefficients) and derived mean diel profiles of $k_{OTG}^{NO_3}$ and $k_{VOC}^{NO_3}$ for the whole campaign (hour averages). The results are shown in Fig. 5, where the red shaded area represents total uncertainty and variability in $k_{VOC}^{NO_3}$ and the black error bars are the total uncertainty in $k_{OTG}^{NO_3}$. Clearly, within combined uncertainty the data overlap, so that missing reactivity is not significantly distinct from zero.

## 3.3 $NO_3$ measurements and comparison with stationary state calculations

Although rough estimates of $NO_3$ concentrations at the Hohenpeissenberg have been made (Handisides et al., 2003; Bartenbach et al., 2007), no direct $NO_3$ measurement had been previously made. For this reason, on just one night during the intensive ($2^{nd}$- $3^{rd}$ of August), the instrument was modified to enable measurement of ambient $NO_3$ mixing ratios rather than $NO_3$-reactivity. The $NO_3$, $O_3$ and $NO_2$ mixing ratios and meteorological data are plotted in Fig. 6.

$NO_3$ mixing ratios slowly increased in the first half of the night, reaching a maximum of 13 pptv around 21:40 UTC. At this time the $O_3$ mixing ratios were also largest and highly variable. After $\approx 22:30$, $O_3$ was slowly removed, the $NO_3$ decreased by a factor of 10 or more indicating that we were sampling more reactive, boundary layer air. This is also evident in the increase in relative humidity and decrease in the temperature until about 01:30.

Given sufficient time, stationary state can be reached for $NO_3$ at night in which the production and loss terms are approximately balanced (Brown et al., 2003a; Crowley et al., 2010; Crowley et al., 2011). In this case $NO_3$ mixing ratios can be described by the ratio of their production rate and loss rate (Eq. 3).

$$[NO_3]_{ss} = \frac{NO_3 \text{ production rate}}{NO_3 \text{ loss rate}} \tag{3}$$

The production rate is governed by the $[NO_2]$ and $[O_3]$ mixing ratio and the corresponding rate constant $k_2$. If the loss processes are due to reaction with VOCs only, this expression becomes:

$$[NO_3]_{ss} = \frac{[O_3][NO_2]k_2}{k_{VOC}^{NO_3}} \tag{4}$$



During this night $k_{\text{OTG}}^{\text{NO}_3}$ was not measured so $k_{\text{VOC}}^{\text{NO}_3}$ was used to account for NO$_3$ losses.

Figure 7 shows the measured NO$_3$ mixing ratios (black) compared to those derived from Eq. (4) using the measured VOC concentrations (red curve). Clearly, the predicted, stationary-state NO$_3$ concentrations are too high (by a factor of up to 3-4), implying that other NO$_3$ loss processes must be considered. As the directly measured reactivity $k_{\text{OTG}}^{\text{NO}_3}$ agrees rather well with

that derived from VOC measurements ($k_{\text{VOC}}^{\text{NO}_3}$) calculated on other campaign nights, it would seem unlikely that unmeasured VOCs contribute sufficiently to NO$_3$ losses to explain this large factor. Stationary-state concentrations of NO$_3$ are influenced not only by VOCs but also by NO (if present at night) and also indirectly via heterogeneous loss of N$_2$O$_5$. Equation (5) can be extended to include these processes (Martinez et al., 2000; Geyer et al., 2001; Brown et al., 2003a; Brown et al., 2003b; Brown et al., 2009; Crowley et al., 2010; Sobanski et al., 2016).

$$[NO_3]_{ss} = \frac{[O_3][NO_2]k_1}{k_{\text{VOC}}^{\text{NO}_3}+k_2[NO]+K_5\,[NO_2]f_{\text{het}}} \qquad (5)$$

where K$_5$ is the equilibrium constant for the forward and reverse reactions (R4, R5). The loss frequency due to heterogeneous uptake of N$_2$O$_5$ to particles ($f_{\text{het}}$) can be calculated by equation 6:

$$f_{\text{het}} \approx \frac{\gamma\,\bar{c}\,A}{4} \qquad (6)$$

which is approximately valid if the particles are less than $\approx$ 1 µm in diameter. In this expression, $A$ is the aerosol surface area

density (cm$^2$ cm$^{-3}$), $\bar{c}$ is the mean, molecular velocity of N$_2$O$_5$ (26233 cm s$^{-1}$ at 298 K) and $\gamma$ is the dimensionless uptake coefficient. If we assume a large value for the uptake coefficient of 0.03 as characteristic for aerosol with low organic content (Bertram and Thornton, 2009; Bertram et al., 2009; Crowley et al., 2011; Phillips et al., 2016) and use the aerosol surface of 1.25-1.55x10$^{-6}$ cm$^2$ cm$^{-3}$ (measured by a scanning mobility particle sizer for 10 – 890 nm), we obtain values for $f_{\text{het}}$ of 2.4-2.9 $\times$ 10$^{-4}$ s$^{-1}$. In this case an unrealistic value of $\gamma$ = 0.5 would be required to lower the calculated, stationary-state

NO$_3$ mixing ratio to between 3 and 10 pptv as observed.

Clearly, heterogeneous losses of N$_2$O$_5$ do not account for the missing NO$_3$ sinks during this night and we now consider the role of NO. As NO mixing ratios in this night did not exceed the detection limit (11 pptv) we used a constant value of 5 pptv to approximately align the calculated NO$_3$ mixing ratio with that measured for much of the night. Clearly, the calculation of NO$_3$-reactivity from stationary-state calculations can be precarious and subject to large cumulative uncertainty from e.g.

measurement uncertainty in NO$_3$ mixing ratios, uptake coefficients, aerosol surface area and NO mixing ratios close to instrumental detection limits.

To assess the NO$_3$ mixing ratios for the rest of the intensive, equation 6 can be augmented by adding the loss rate constant for NO$_3$-photolysis $J_{\text{NO}_3}$ and substituting $k_{\text{VOC}}^{\text{NO}_3}$ for $k_{\text{OTG}}^{\text{NO}_3}$.

$$[NO_3]_{ss} = \frac{[O_3][NO_2]k_1}{k_{\text{OTG}}^{\text{NO}_3}+k_2[NO]+J_{\text{NO3}}+K_5\,[NO_2]f_{\text{het}}} \qquad (7)$$

In the absence of a direct measurement of $J_{\text{NO}_3}$, the diel cycle of the relative NO$_3$-to-NO$_2$ photolysis rate constant ($J_{\text{NO}_3}/J_{\text{NO}_2}$) was            calculated            using            the            TUV            (tropospheric            ultraviolet            and            visible            radiation)            model





(https://www2.acom.ucar.edu/modeling/tropospheric -ultraviolet-and-visible-tuv-radiation-model) and then put on an absolute basis using measured $J_{NO2}$ values.

Figure 8 shows the $NO_3$ production rate (lower panel, black curve) and total loss rate (lower panel, red curve) as well as the stationary-state $NO_3$ mixing ratios for the entire intensive period (Fig. 8, upper panel black curve). During nights in which the-reactivity fell below the detection limit of the instrument $k_{OTG}^{NO_3}$ was set to 0.005 s$^{-1}$. Calculated $NO_3$ mixing ratios were in the sub-pptv range during daytime and around 1-15 pptv during nighttime. The $NO_3$ mixing ratios thus derived are comparable to those measured on a single night (Fig. 8, red curve) and are broadly consistent with previous estimates for this site (Handisides et al., 2003; Bartenbach et al., 2007).

## 3.4 Contribution to $NO_x$ loss

At nighttime, in the absence of NO and sunlight, each $NO_3$ radical formed in the reaction of $NO_2$ with $O_3$ will either be removed indirectly via the uptake of $N_2O_5$ onto particles or will react with a biogenic hydrocarbon. The latter results in the formation of an organic nitrate at a yield of between 20 and 100%, depending on the specific VOC (Ng et al., 2017). The large daytime values for $k_{OTG}^{NO_3}$ obtained in this study suggest that even during sunlight hours (when $NO_3$ is generally considered to be of little significance owing to its rapid photolysis) significant amounts of $NO_3$ form organic nitrates rather than reforming $NO_2$ by reaction with NO, or photolysis.

The fraction, $f$, of $NO_3$ that will react with organic trace gases is given by:

$$f = \frac{k_{OTG}^{NO_3}}{\left(\left[k_{OTG}^{NO_3}\right]+\left[J_{NO3}\right]+[NO]k_3+K_5[NO_2]f_{het}\right)} \tag{8}$$

where the denominator sums all loss processes for $NO_3$. Figure 9 illustrates this via a diel cycle of the median for $f$. At nighttime, $\approx$ 99% of the $NO_3$ will be lost to reaction with BVOCs, with indirect heterogeneous losses representing the remaining 1%. During daytime, at the peak of the actinic flux (max $J_{NO3} \approx$ 0.2 s$^{-1}$) and correspondingly high levels of NO ($k_{NO}$ = 0.1 - 0.2 s$^{-1}$), 20% of the formed $NO_3$ was lost due to reaction with organic trace gases, increasing up to 40% in the late afternoon. This result is comparable with reactivity measurements in a boreal forest in Finland during IBAIRN 2016 where a very similar diel profile for $f$ was determined (Liebmann et al., 2018). The $NO_3$-reactivity data from these measurements indicate that the role of $NO_3$ as a daytime oxidant of biogenic VOCs in forested regions may so far have been underestimated, which in turn has implications for understanding the diel cycle of organic nitrate and secondary organic aerosol formation in such environments.

## 4 Comparison of $NO_3$ and OH-reactivity

As mentioned above $NO_3$ radicals and OH radicals react with atmospheric trace gases via different mechanisms, resulting in profoundly different rate coefficients and thus reactivities. By combining the continuous, on-site measurements of the OH-





reactivity with the NO$_3$-reactivity measurements during the intensive period, we were able to generate the first dataset of simultaneous, direct measurement of both OH-reactivity, $k_{total}^{OH}$, and NO$_3$-reactivity at any location.

To aid comparison, we subtracted the contributions of several inorganic and organic trace gases (NO, NO$_2$, SO$_2$, CO, CH$_4$) that are not included in $k_{OTG}^{NO_3}$ or do not react to a significant extent with NO$_3$ from the total OH-reactivity and thus derived $k_{OTG}^{OH}$.

$$k_{OTG}^{OH} = k_{total}^{OH} - k_{NO}^{OH}[NO] - k_{NO_2}^{OH}[NO_2] - k_{SO_2}^{OH}[SO_2] - k_{CH_4}^{OH}[CH_4] - k_{CO}^{OH}[CO] \qquad (9)$$

Fig. 10 depicts the time series of $k_{Total}^{OH}$, $k_{OTG}^{OH}$ and $k_{OTG}^{NO_3}$. All display maximum values close to midday, though $k_{OTG}^{OH}$ averaged over the intensive are larger by a factor of 44 larger than $k_{OTG}^{NO_3}$, reflecting generally larger rate coefficients for OH. The blue shaded areas for $k_{Total}^{OH}$ represents the 1 σ uncertainty of the measurements. Total uncertainty in $k_{OTG}^{OH}$ and $k_{OTG}^{NO_3}$ is not shown to preserve clarity of presentation. The time series of the $k_{OTG}^{OH}$ can be found in the supplementary information (Fig. S6).

The measured reactivities of both radicals show a clear diel profile, with higher daytime and lower nighttime values. Figure 11 shows a correlation plot of OH and NO$_3$-reactivity divided into day (red) and nighttime (black) data. During the day, the data are highly scattered, which can be understood when one considers the highly variable organic content of the air masses being sampled. To illustrate this we have drawn the expected correlation lines (based on the known, relative rate coefficients) for single component organic trace gases including isoprene and terpenes. The expected slopes for these individual VOCs are very different and encompass the full scatter in the observations, which is the result of changing atmospheric composition (i.e. the mix of reactive organic species) owing to changes in air mass age and source region (wind direction) during the campaign. The extremes are represented by α-terpinene (which favours NO$_3$) and CH$_4$ (which favours OH).

During nighttime (black points) the plot of $k_{OTG}^{OH}$ versus $k_{OTG}^{NO_3}$ is less scattered, indicating that the air masses sampled (often from the residual layer) are chemically less complex and variable. The data lay close to the line that marks the expected correlation if isoprene were the dominant sink of both NO$_3$ and OH at nighttime once molecules such as CO and CH$_4$ have been removed from the term describing OH-reactivity. This is in broad agreement with our observation that isoprene is the main sink of NO$_3$ during nights when the measurement site was decoupled from direct boundary layer emissions.

## 5. Summary and Conclusion

Direct measurements of the NO$_3$-reactivity towards organic trace gases, $k_{OTG}^{NO_3}$ were conducted at the top of the Hohenpeissenberg mountain (988 m a.s.l.) during an intensive measurement campaign in the summer of 2017. NO$_3$-reactivities had a distinct diel profile with values as large as 0.3 s$^{-1}$ during daytime but close to or below the detection limit of the instrument during nighttime when the measurement site was frequently in the residual layer / free troposphere. Within experimental uncertainty, the high daytime NO$_3$-reactivity was accounted for by BVOCs that were measured at the site, and



was dominated by monoterpenes especially α-pinene and sabinene. On average, the reaction with VOCs accounted for ≈ 99% of the loss of $NO_3$ during nighttime and an average of 20% at noon, increasing to 30-50% during early morning and late evening. The reaction of $NO_3$ with BVOCS therefore represents a significant $NO_x$ loss not only during the night but also during daytime and implies significant formation of organic nitrates via $NO_3$ reactions throughout the diel cycle. Stationary-state, daytime and nighttime $NO_3$ mixing ratios were calculated using the production term and $k_{OTG}^{NO_3}$ and were broadly consistent with direct measurement made on one night. A comparison between directly measured OH- and $NO_3$ reactivities was performed, indicating a weak correlation during the day when chemically reactive, complex and variable air masses were encountered. A tighter correlation, consistent with isoprene dominating the (low) $NO_3$-reactivities was observed at night.

**Acknowledgements:** We would like to thank the DWD for hosting and supporting this measurement campaign. We would like to thank the DWD personnel for the data evaluation, most notably Katja Michl for NMHCs, Jennifer Englert for OVOCs, Harald Flentje for the SMPS data as well as the great technical support, particularly we are grateful to Erasmus Tensing, Thomas Elste, Georg Stange. We thank Chemours for provision of the FEP sample used to coat the CRD-cavities.



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



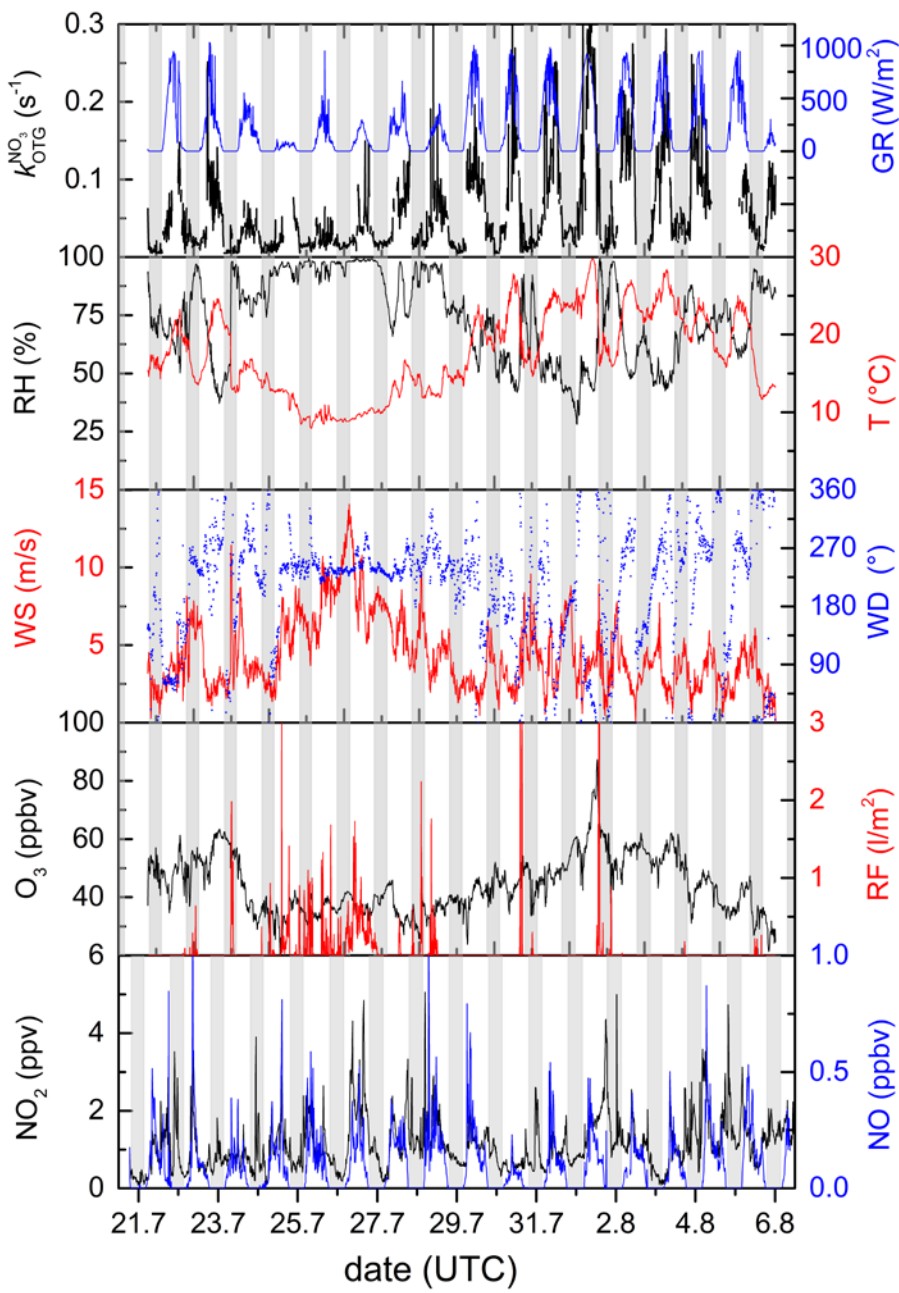

Figure 1: Overview of measurements during the 2017-intensive. The grey shaded area represents nighttime. GR = global radiation, RF = rainfall, RH = relative humidity, T = temperature, WS = wind speed, WD = wind direction.



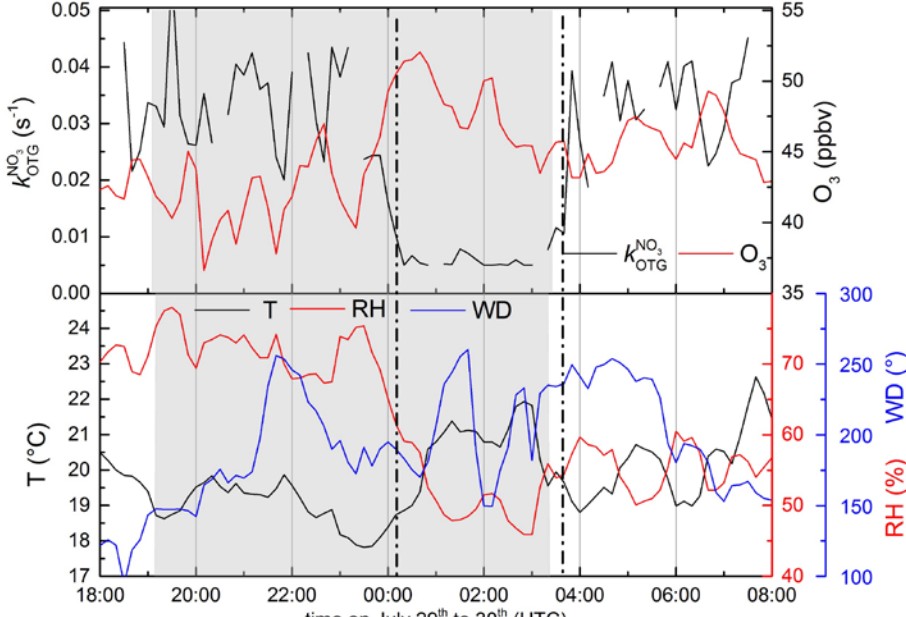

Figure 2: Upper panel: $k_{OTG}^{NO_3}$ (black) and $O_3$ mixing ratios (red) from the 29$^{th}$ to the 30$^{th}$ of July. From 23:50 UTC until sunrise the measurement site is located in the residual layer / free troposphere. Lower panel: temperature (T), relative humidity (RH) and wind direction (WD) during the same period.





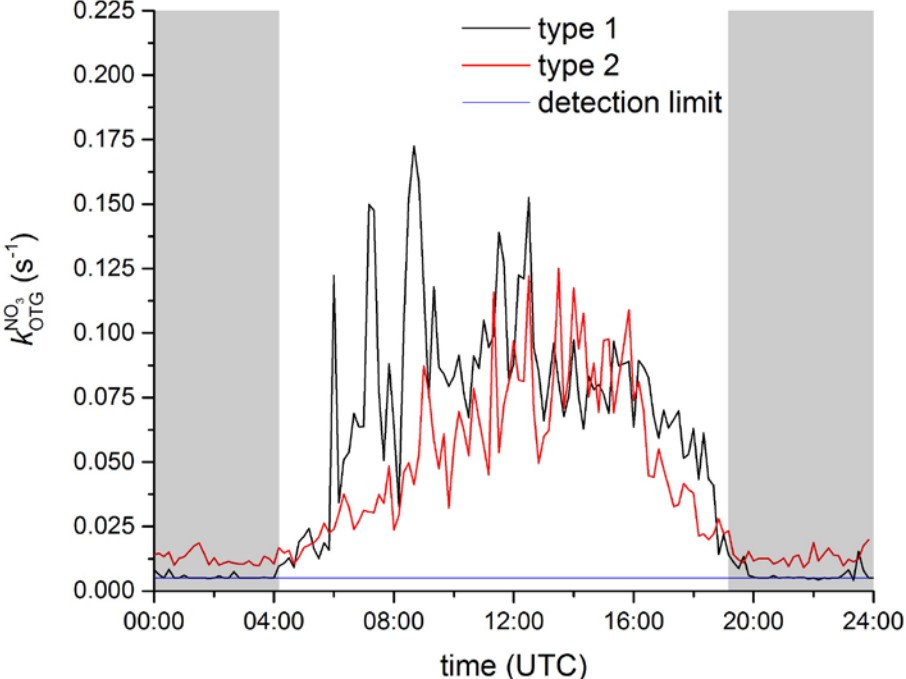

Figure 3: Median, diel profile of the $NO_3$-reactivity. Type 1 nights (black line) show values around the detection limit during night, type 2 nights (red line) are above the detection limit.





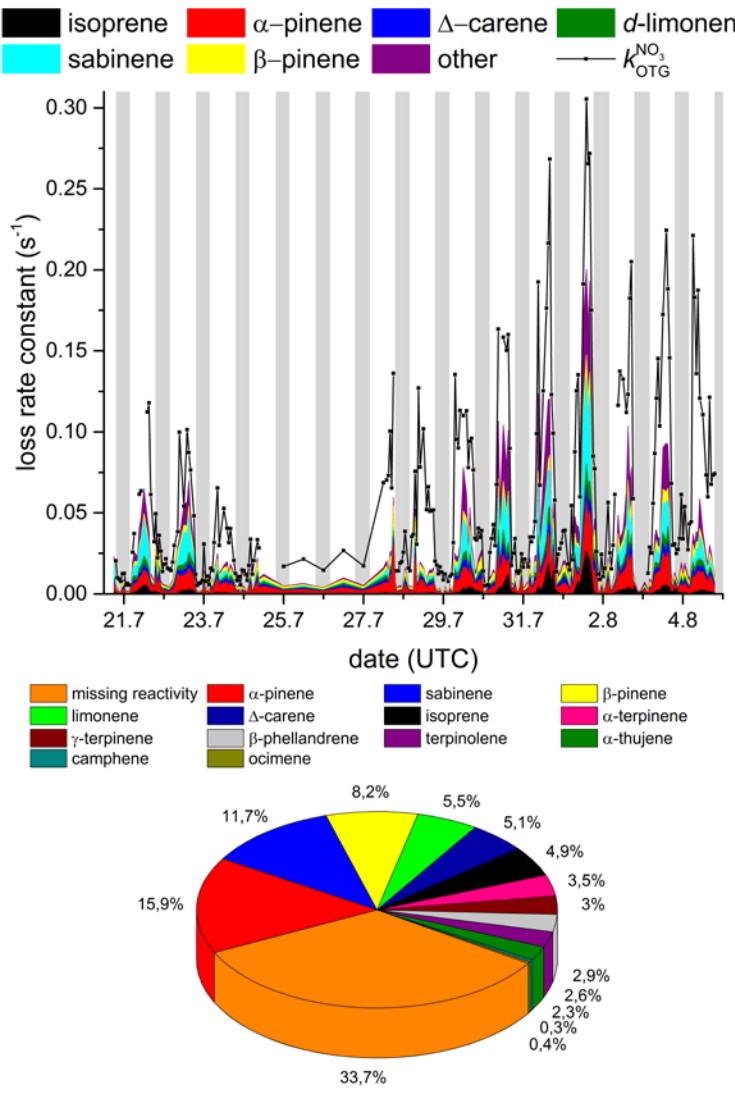

Figure 4: Upper panel: measured values of $k_{OTG}^{NO_3}$ (black) in comparison with the loss rate constant assigned to individual VOCs. The term "other" includes terpinolene, β-phellandrene, α-terpinene, γ-terpinene, α-thujene and camphene. Myrcene and α-phellandrene were also measured but below the detection limit during the whole campaign. The lower panel indicates the campaign averaged contribution of each measured VOC to the $NO_3$ loss rate as well as reactivity that was not accounted for by measured VOCs ("missing reactivity").





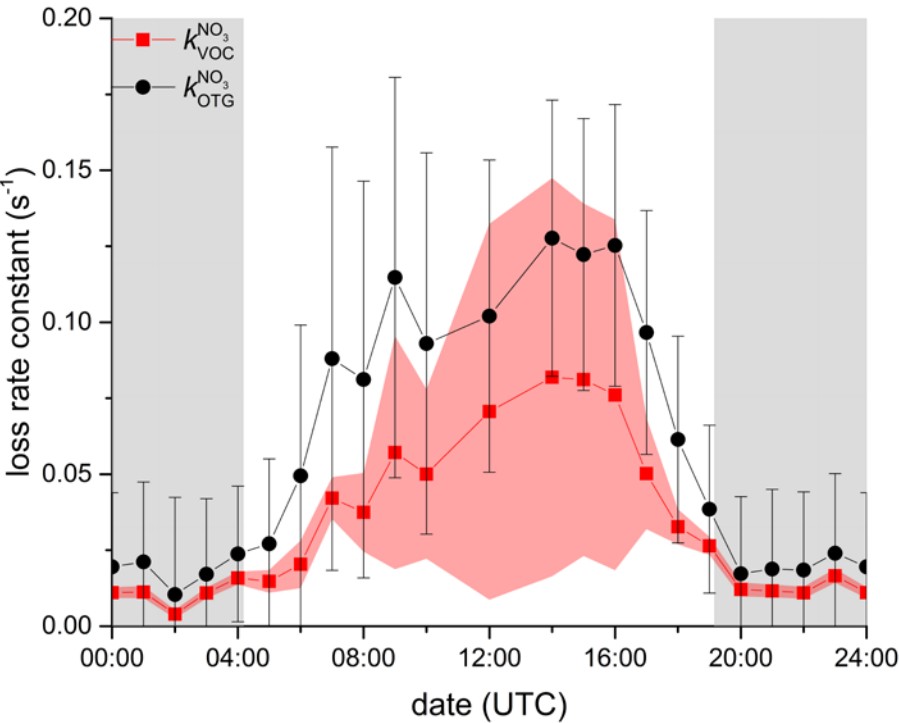

Figure 5: Median diel profiles of $k_{VOC}^{NO_3}$ and $k_{OTG}^{NO_3}$. The error bars on the $k_{OTG}^{NO_3}$ measurements are total uncertainty, including systematic error and variability. The uncertainty in $k_{VOC}^{NO_3}$ (shaded red area) is dominated by uncertainty in the mixing ratios of the VOCs.



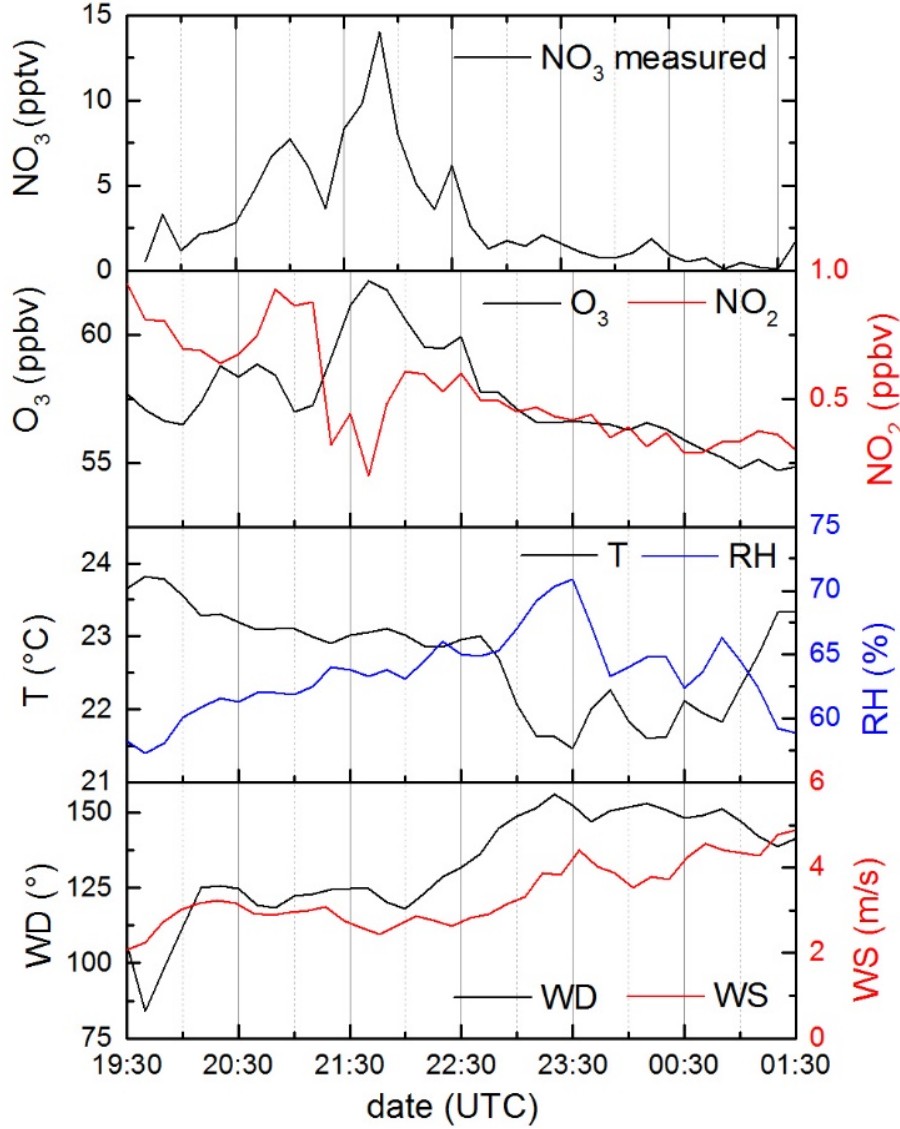

Figure 6: $NO_3$ mixing ratios measured in the night from the 2th – 3rd of August as well as $NO_2$ and $O_3$ (which define the $NO_3$ production rate). T = temperature, RH = relative humidity, WD = wind directions, WS = wind speed.



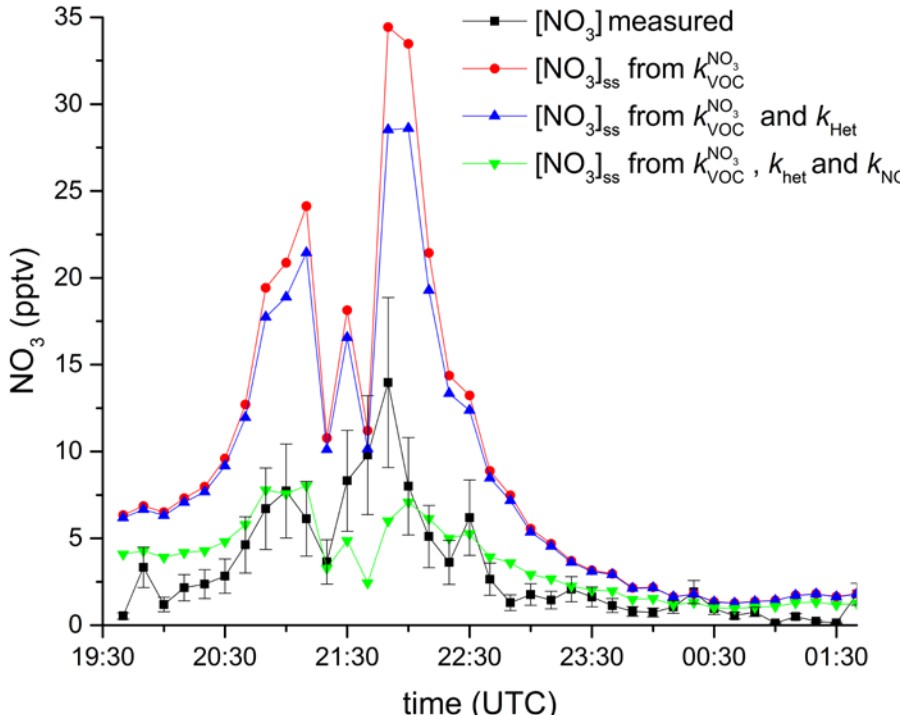

Figure 7: Comparison of measured $NO_3$ mixing ratio (black) with calculated stationary-state mixing ratios using $k_{VOC}^{NO_3}$ (red), $k_{VOC}^{NO_3} + k_{Het}$ (blue), and $k_{VOC}^{NO_3}, + k_{Het} + k_{NO}$ (green).



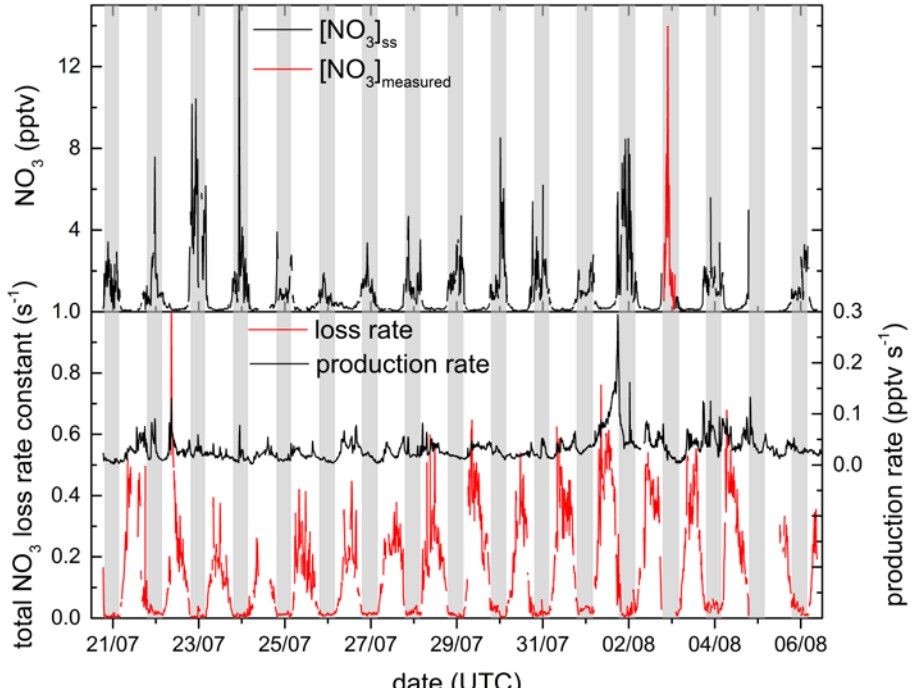

Figure 8: Upper panel: Stationary-state $NO_3$ mixing ratios calculated using $k_{OTG}^{NO_3}$, $[NO]k_3$, $K_5[NO_2]f_{het}$ and $J_{NO_3}$ for the entire campaign and comparison with the measured $NO_3$ mixing ratios (03.08). The lower panel plots the time series of production and loss rates used for calculation of $[NO_3]_{ss}$.



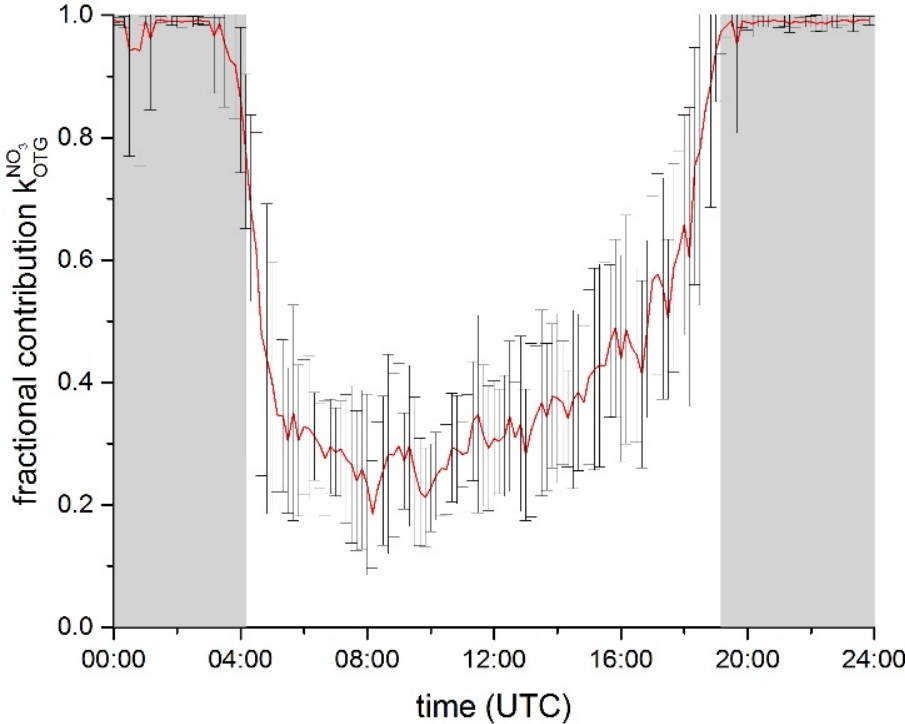

Figure 9: The fraction, $f$, of the total $NO_3$ loss with organic trace gases as a campaign mean, diel cycle where $f =$ $k_{OTG}^{NO_3} / (k_{OTG}^{NO_3} + J_{NO_3} + [NO]k_3 + K_5[NO_2]f_{het})$. The error bars reflect variability only and do not consider systematic uncertainty.



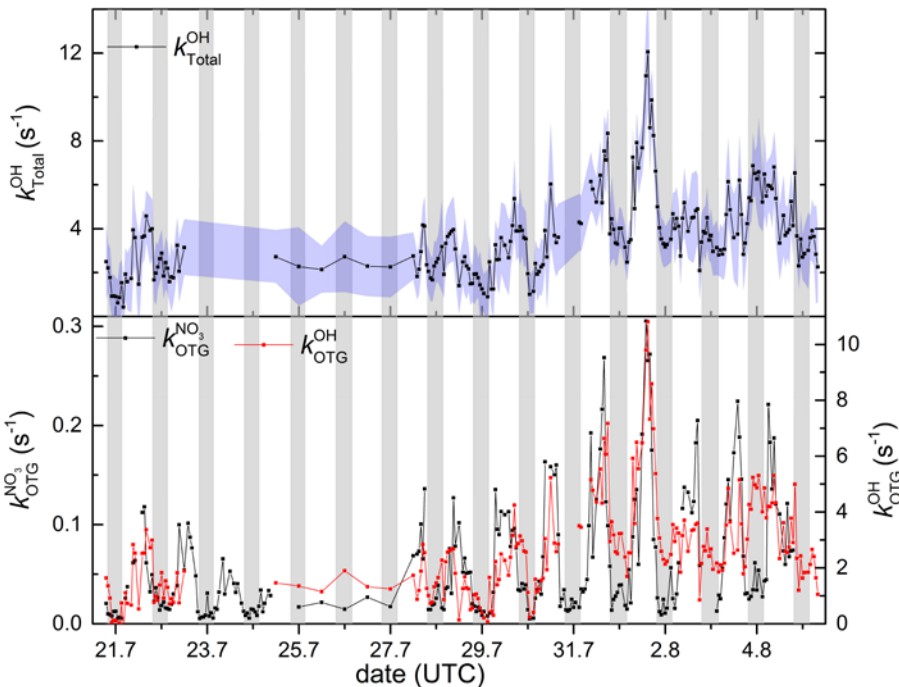

Figure 10: Upper panel: Time series of $k_{Total}^{OH}$ (shaded region is 1σ uncertainty). Lower panel: Time series of $k_{OTG}^{NO_3}$ and $k_{OTG}^{OH}$. The data is plotted so that the curves overlay at the peak reactivity (01/08).





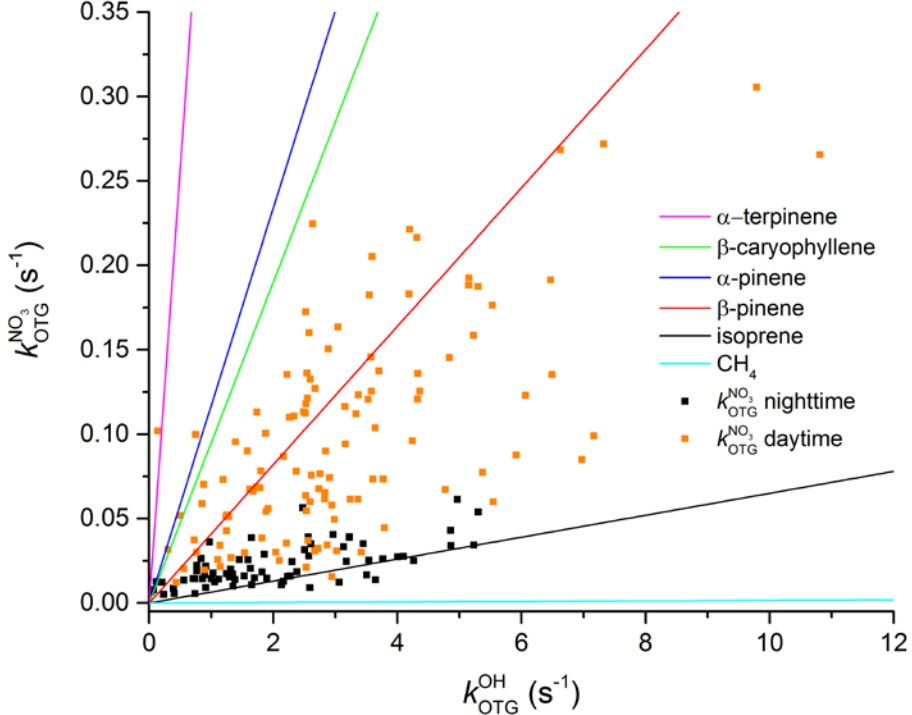

Figure 11: Correlation between OH-reactivity and NO$_3$-reactivity. The coloured lines are relative NO$_3$ and OH reactivity for single VOCs. The measured NO$_3$ and OH reactivities are depicted as black (nighttime) and orange datapoints (daytime).