# Peer review of "Direct measurements of $\text{NO}_3$ -reactivity in and above the boundary layer of a mountain-top site: Identification of reactive trace gases and comparison with OH-reactivity."

_Atmospheric Chemistry and Physics, 2018_

## Referee Comment (RC1) · Anonymous Referee #1 · 11 May 2018

**Direct measurements of NO$_3$-reactivity in and above the boundary layer of a mountain-top site: Identification of reactive trace gases and comparison with OH-reactivity**

**Journal:** Atmospheric Chemistry and Physics
**Manuscript ID:** acp-2018-324

The manuscript by Jonathan M. Liebmann et al. describes direct measurements of NO$_3$ and OH reactivity at a mountain site in Germany during the late July to early August period, 2017. The reactivity measurements are supported by other instrumentation such as GC-MS/FID for VOC speciation and quantification and NO$_2$, NO, and O$_3$ data. The diurnal profile of NO$_3$ reactivity towards organic trace gases, k$_{OTG,}$ shows low nighttime and relatively higher daytime values as a result of the measurement site being decoupled by surface emissions for most nights. The measured k$_{OTG}$ is in agreement with the calculated one using monoterpene and isoprene measurements, within the measurement uncertainty. As in previous work from the same group, the daytime NO$_3$ reactivity shows significant contribution from the reaction of NO$_3$ with biogenic VOCs (BVOCs) supporting the BVOC role to NO$_x$ removal and the formation of organic nitrates at the measurement site. Finally, this work is the first to provide simultaneous NO$_3$ and OH reactivity measurements which enables the authors to explore the correlation between the two radical chemistries.

The manuscript is well written and the results and data interpretation is clearly presented. This work expands further the field observations on NO$_3$ reactivity and for the first time enables a direct comparison with OH reactivity measurements. Therefore, I recommend the article's publication in ACP. A few points for consideration are given below:

* page 5, line 28-29 and page 6, line 5: I would suggest to add the magnitude/range of the corrections

* page 6, line 24: adding supplemental information or a reference for the transmission experiments would be useful

* Section 3.2, Figures 5: Did the authors compare measured and calculated NO$_3$ reactivity values for type 2 nights only, where BVOCs are the major contributors? I am wondering whether there is a systematic difference in reactivity due to "unknown" VOCs and not just random uncertainty in the measurements.

* page 12-13, section 4: The authors should discuss, even briefly, the agreement between measured and calculated OH reactivity as shown in Figure S6. Is there any indication of missing OH reactivity? This comparison could provide more insight on "unknown" VOCs that might explain a possible systematic difference in both OH and NO$_3$ reactivity. For instance, would a plot of "missing" reactivities, dk$_{OH}$ vs. dk$_{NO3}$, be informative?

**Other revisions / typos:**

* page 1, line 16: I believe it should be "high reactivity" instead of "reactivity high" and "but" should be replaced with "and"

* page 4, line 1: replace "outer-diameter" with "O.D."

* page 4, line 2: is this flow at STP conditions?

* page 4, line 13:  should be "thermostated"

* page 6, line 24: shouldn't this be "NO addition" ?

* page 7, line 20: sentence should read ".. for 15 min ($C_2$-$C_8$) and 20 min ($C_5$-$C_{13}$), respectively .."

* page 7, line 28: spaces are needed in "$ms^{-1}$"

* page 15, line 19: should be "GCxGC-FID"

* Figure 1: Fix $NO_2$ units

---

## Referee Comment (RC2) · Anonymous Referee #2 · 11 Jun 2018

The manuscript Liebmann et al. presents NO3 and OH reactivity measurements at a forested mountaintop site in Germany. The data is compared to reactivity calculations based on individual VOC reactions and heterogeneous uptake. The data is also discussed with respect to meteorological conditions and their impact on the VOC sources.

Overall, the study present interesting and high-quality data, but it seems that the take-home message of the manuscript is not entirely clear. The most innovative part is that of the error calculation of the speciated reactivity calculations and the question of the missing NO3 sink. As I will outline below this section needs to be expanded to more

clearly support its main conclusion. The comparison of NO3 and OH reactivities is interesting, but I am not entirely clear what the significance of the comparison is (except for the question of the VOC mix, which could as easily be addressed through the VOC observations). Maybe it would be useful to add the speciated reactivity calculation for OH to determine which of the two radical chemistries is less certain. Aside from these possible improvements the manuscript is well written and with a few revision, following my comments below, is suitable for publication in ACP.

Comments:

Section 2: Please add a subsection on the aerosol size measurements used later in the manuscript.

Page 5 line 15 - 16: Is this number in meters? Please add unit

Section 3.2: As mentioned before I applaud the authors for considering the uncertainty of the speciated reactivity calculation. However, this discussion is really confusing and I have a hard time following the statements made in this section. The fit in Figure S5 was performed using errors for kˆNO3_VOC, thus the uncertainty of the slope should be a statistical measure of the uncertainty of the missing reactivity. The manuscript does currently not explain where these errors come from. If they are the same errors as used for Fig 5, then the fit I Fig S5 seems to show a statistically significant missing reactivity. Fig 5 on the other hand seem to show that there is no statistically significant missing reactivity. This needs to be discussed in more detail. It seems to me that the fit in Fig S5 is a statistically more meaningful measure, and I would recommend moving this figure to the main manuscript. I also think it would benefit the scientific value of the manuscript if a statement on the main causes of the uncertainty and future steps to reduce them could be added.

Figure 4: Please add some measure for uncertainty to this figure and/or provide a more detailed discussion in the caption.

Page 10, line 10: Why assume a 15% uncertainty and not use published uncertainties?

Section 3.3 vs Section 3.4: Unless I misunderstand the arguments in these two sections, they seem contradictory for the nighttime data. In section 3.3 and Figure 7 NO is needed at night to close the NO3 budget. It appears that NO is responsible for around 50% of the NO3 loss. However, in Section 3.4 NO seems to have been ignored at night, therefore leading to the 99% fractional contribution of NO3 + organics. So which one is correct? Is there sufficient NO at night to destroy NO3 or not?

Section 3.4: Should the title be "Contribution to NO3 loss"? I don't see the NOx loss discussed anywhere.
* * *

---

## Author Comment (AC1) · 4 Jul 2018

In the following, the referee's comments are reproduced (black) along with our replies (blue) and changes made to the text (red) in the revised manuscript.

**Referee 1**

The manuscript by Jonathan M. Liebmann et al. describes direct measurements of NO3 and OH reactivity at a mountain site in Germany during the late July to early August period, 2017. The reactivity measurements are supported by other instrumentation such as GC-MS/FID for VOC speciation and quantification and NO2, NO, and O3 data. The diurnal profile of NO3 reactivity towards organic trace gases, kOTG, shows low nighttime and relatively higher daytime values as a result of the measurement site being decoupled by surface emissions for most nights. The measured $k_{OTG}$ is in agreement with the calculated one using monoterpene and isoprene measurements, within the measurement uncertainty. As in previous work from the same group, the daytime NO3 reactivity shows significant contribution from the reaction of NO3 with biogenic VOCs (BVOCs) supporting the BVOC role to NOx removal and the formation of organic nitrates at the measurement site. Finally, this work is the first to provide simultaneous NO3 and OH reactivity measurements which enables the authors to explore the correlation between the two radical chemistries.

The manuscript is well written and the results and data interpretation is clearly presented. This work expands further the field observations on NO3 reactivity and for the first time enables a direct comparison with OH reactivity measurements. Therefore, I recommend the article's publication in ACP.

We thank referee 1 for this review and overall positive assessment of our manuscript. The manuscript has been improved in line with the comments listed below.

A few points for consideration are given below:

page 5, line 28-29 and page 6, line 5: I would suggest to add the magnitude/range of the corrections

Corrected, we now write:

Corrections were applied to take into account NO loss ($\approx$ 5-30%) and $NO_2$ formation ($\approx$ 0 - 12%, typically $\approx$ 2%) due to further reactions involving ozone in the inlet tubing.

page 6, line 24: adding supplemental information or a reference for the transmission experiments would be useful.

We have corrected it to:

The $NO_3$ transmission through the inlet (67 $\pm$ 15%), filter and filter-holder (84 $\pm$ 10%) and cavity (88 $\pm$ 10%) were established post-campaign as described by Schuster et al. (2009) and used to correct the data.

Section 3.2, Figures 5: Did the authors compare measured and calculated NO3 reactivity values for type 2 nights only, where BVOCs are the major contributors? I am wondering whether there is a systematic difference in reactivity due to "unknown" VOCs and not just random uncertainty in the measurements.

As we state in the manuscript, there is no significant difference between calculated reactivity (via VOCs) and that measured. There is no evidence for "unknown" VOCs.

page 12-13, section 4: The authors should discuss, even briefly, the agreement between measured and calculated OH reactivity as shown in Figure S6. Is there any indication of missing OH reactivity? This comparison could provide more insight on "unknown" VOCs that might explain a possible systematic difference in both OH and NO3 reactivity.

The measured OH reactivity and its comparison to that calculated from VOC mixing ratios is not the central theme of this manuscript. Nonetheless, we now indicate that the measured values are larger than calculated, and indicate that for some periods of the campaign there is evidence for unattributed reactivity for OH. We now write:

The time series of $k_{OTG}^{OH}$ can be found in the supplementary information (Fig. S6) where it is compared to that calculated from individual VOCs at the corresponding rate coefficient, $k_{VOC}^{OH}$. As often the case for OH, there are periods in which the calculated reactivity significantly underestimates (up to a factor of $\approx$ four) the measured values. A detailed comparison of measured and calculated OH-reactivity at this location and including data over a much longer time period will be subject of future publications.

For instance, would a plot of "missing" reactivities, dkOH vs. dkNO3, be informative?

As described in the manuscript, there is no significant difference in the measured and calculated $NO_3$ reactivity, i.e. there is no evidence for "missing reactivity". A plot of "missing" OH reactivity versus "missing" $NO_3$ reactivity is therefore not warranted.

**Other revisions / typos:**

page 1, line 16: I believe it should be "high reactivity" instead of "reactivity high" and "but" should be replaced with "and"

Corrected, we now write:

The diel cycle of $k_{OTG}^{NO_3}$ was strongly influenced by local meteorology with high reactivity observed during the day (values of up to 0.3 s$^{-1}$) and values close to the detection limit (0.005 s$^{-1}$) at night when the measurement site was in the residual layer / free troposphere.

page 4, line 1: replace "outer-diameter" with "O.D."

We prefer to keep outer-diameter as it is only used twice in the whole document

page 4, line 2: is this flow at STP conditions?

Yes. Correction made.

page 4, line 13: should be "thermostated"

We believe that "thermostatted" is the correct spelling.

page 6, line 24: shouldn't this be "NO addition" ?

Corrected, we now write:

(NO addition)

page 7, line 20: sentence should read ".. for 15 min (C2-C8) and 20 min (C5-C13), respectively .."

We now write:

VOCs were sampled every hour for either 15 ($C_2$–$C_8$) or 20 mins ($C_5$–$C_{13}$).

page 7, line 28: spaces are needed in "ms-1"

Corrected, we now write:

Wind speeds were generally around 2.5 to 7.5 m s$^{-1}$, increasing up to 15 m s$^{-1}$ during the rainy periods.

page 15, line 19: should be "GCxGC-FID"

Corrected

Figure 1: Fix NO2 units

Corrected

**Referee 2**

The manuscript Liebmann et al. presents NO3 and OH reactivity measurements at a forested mountaintop site in Germany. The data is compared to reactivity calculations based on individual VOC reactions and heterogeneous uptake. The data is also discussed with respect to meteorological conditions and their impact on the VOC sources. Overall, the study present interesting and high-quality data, but it seems that the takehome message of the manuscript is not entirely clear. The most innovative part is that of the error calculation of the speciated reactivity calculations and the question of the missing NO3 sink. As I will outline below this section needs to be expanded to more clearly support its main conclusion. The comparison of NO3 and OH reactivities is interesting, but I am not entirely clear what the significance of the comparison is (except for the question of the VOC mix, which could as easily be addressed through the VOC observations). Maybe it would be useful to add the speciated reactivity calculation for OH to determine which of the two radical chemistries is less certain. Aside from these possible improvements the manuscript is well written and with a few revision, following my comments below, is suitable for publication in ACP.

We thank referee 2 for this review and positive assessment of our manuscript. The manuscript has been improved in line with the comments listed below.

Section 2: Please add a subsection on the aerosol size measurements used later in the manuscript.
We have added the following section:

**2.6 Particle measurements**

The aerosol surface area was calculated using particle number size distributions obtained from a custom-built SMPS described in detail in Wiedensohler et al. (2012) and Birmili et al. (2016). Briefly, the instrument uses a Vienna-type DMA with a condensation particle counter (CPC model 3772, TSI Inc.) to measure particles between 10 and 800 nm. The sheath flow rate is 5 L min$^{-1}$ at an aerosol flow rate of 1 L min$^{-1}$, both are actively dried. The typical time resolution for one combined up-scan and down-scan is 5 min.

Page 5 line 15 - 16: Is this number in meters? Please add unit
This number does not have any unit as it is the ratio of the length of the cavity filled with absorber divided by the length of the cavity.

Section 3.2: As mentioned before I applaud the authors for considering the uncertainty of the speciated reactivity calculation. However, this discussion is really confusing and I have a hard time following the statements made in this section. The fit in Figure S5 was performed using errors for kˆNO3_VOC, thus the uncertainty of the slope should be a statistical measure of the uncertainty of the missing reactivity. The manuscript does currently not explain where these errors come from.
The standard deviation (≈ 0.15) previously reported for the slope (1.5) is based on a statistical fit to the data and depends on the number of data points sampled. However, the uncertainty (both x and y) stems mainly from systematic error which is not reduced by sampling large datasets. i.e. the uncertainty on the

slope does not show that the "missing" reactivity is significant. We have added a time series of data to Fig. 5 to illustrate this.

In the caption to Figure S5 we now write:

The error bars shown for $k_{VOC}^{NO_3}$ were derived from uncertainties in the mixing ratios of the VOCs and the rate constant for the reaction with $NO_3$ as described in Section 3.2. The error bars shown for $k_{OTG}^{NO_3}$ were calculated as described in the text.

If they are the same errors as used for Fig 5, then the fit I Fig S5 seems to show a statistically significant missing reactivity. Fig 5 on the other hand seem to show that there is no statistically significant missing reactivity.

These are not the same uncertainties as the error-bars on the diel profile (Figure 5) also contains variability, as indicated in the caption.

Mean diel profiles of $k_{VOC}^{NO_3}$ and $k_{OTG}^{NO_3}$. The error bars on the $k_{OTG}^{NO_3}$ measurements are total uncertainty, including systematic error and variability. The uncertainty in $k_{OTG}^{NO_3}$ that derives from the corection procedure (effect of NO and $NO_2$) and dilution accuracy was calculated as described by Liebmann et al (2017). The uncertainty in $k_{VOC}^{NO_3}$ (shaded red area) is dominated by uncertainty in the mixing ratios of the VOCs.

This needs to be discussed in more detail.

We have extended Figure 5 to include a time series of plot of $k_{VOC}^{NO_3}$ and $k_{OTG}^{NO_3}$ with respective (overlapping) uncertainties.

It seems to me that the fit in Fig S5 is a statistically more meaningful measure, and I would recommend moving this figure to the main manuscript.

See above: The standard deviation ($\approx 0.15$) previously reported for the slope (1.5) is based on a statistical fit to the data and depends on the number of data points sampled. However, the uncertainty (both x and y) stems mainly from systematic error which is not reduced by sampling large datasets. i.e. the uncertainty on the slope does not show that the "missing" reactivity is significant.

I also think it would benefit the scientific value of the manuscript if a statement on the main causes of the uncertainty and future steps to reduce them could be added.

We have added a statement:

The comparison of measured $NO_3$ reactivity with that calculated from VOC measurements suffers from substantial uncertainties in both parameters. For this campaign, the uncertainty in $NO_3$ reactivity was larger than previously reported (Liebmann et al., 2018) due to reduced stability of the $NO_3$ source, which will be improved in future versions of the instrument. A further obstacle to calculation of unattributed reactivity was the uncertainty in the measurements of biogenic VOCs and the associated rate constants for the $NO_3$ reaction. The latter could be reduced by more, accurate data on $NO_3$ rate coefficients with some of the more important biogenic species.

Figure 4: Please add some measure for uncertainty to this figure and/or provide a more detailed discussion in the caption.

Addition of uncertainty (e.g. in the contributions of individual VOCS) would render this Figure illegible. Instead we have added in Figure 5 a time series of $k_{\mathrm{VOC}}^{\mathrm{NO_3}}$ and $k_{\mathrm{OTG}}^{\mathrm{NO_3}}$ with respective (overlapping) uncertainties.

Page 10, line 10: Why assume a 15% uncertainty and not use published uncertainties?

We have used IUPAC recommended rate constants. IUPAC indicate that assessment of uncertainties is not based on rigorous, statistical methods as the underlying information is usually not available and proper assessment of systematic error is not always performed in the studies they evaluate. The uncertainties that are listed by IUPAC are conservative and (e.g. if single measurements are available) may be as large as a factor two. Generally, based on consideration of experimental techniques and number of studies, IUPAC does not list uncertainties that are less that about 10-15 % (even when several results agree) and use of IUPAC uncertainties would further increase the uncertainty in $k_{\mathrm{VOC}}^{\mathrm{NO_3}}$. We have added text to indicate that use of IUPAC listed uncertainties would increase the uncertainty in $k_{\mathrm{VOC}}^{\mathrm{NO_3}}$.

We note that IUPAC listed uncertainties for each VOC are generally larger 15 % (especially when only few studies are available) and the use thereof would substantially increase the uncertainty in $k_{\mathrm{VOC}}^{\mathrm{NO_3}}$. The results are shown as a time series and a campaign averaged, diel profile in Fig. 5. For the time series, we plot data points only when values of both $k_{\mathrm{VOC}}^{\mathrm{NO_3}}$ and $k_{\mathrm{OTG}}^{\mathrm{NO_3}}$ were available. The time series illustrates that, within combined uncertainty, the data overlap, so that missing reactivity is not significantly distinct from zero apart from a few isolated data points. This contrasts the conclusions of a previous campaign (boreal forest) with this instrument (Liebmann et al., 2018) in which up to 40-60% of the measured reactivity could not be accounted for by measured VOCs.

Section 3.3 vs Section 3.4: Unless I misunderstand the arguments in these two sections, they seem contradictory for the nighttime data. In section 3.3 and Figure 7 NO is needed at night to close the NO3 budget. It appears that NO is responsible for around 50% of the NO3 loss. However, in Section 3.4 NO seems to have been ignored at night, therefore leading to the 99% fractional contribution of NO3 + organics. So which one is correct? Is there sufficient NO at night to destroy NO3 or not?

The data in Figure 7, were obtained on a night with very low VOC mixing ratios (residual layer) and a very low NO mixing ratio (5 pptv) is sufficient to balance the NO$_3$-budget. As this low mixing ratio is below the instrument's detection limit we do not conclude that NO was responsible, we merely indicate its potential role. The text has been modified:

The efficient reaction of NO with NO$_3$ means that low mixing ratios of NO can contribute to NO$_3$ reactivity, and we calculate that $\approx$ 5 pptv NO would approximately align the calculated NO$_3$ mixing ratio with that measured for much of the night. This value is however below the detection limit of the CLD (11 pptv) used to measure NO and we cannot conclude that NO at such levels was responsible for the reduction in NO$_3$ levels required to bring observation and steady-state calculation into agreement.

Section 3.4. does not ignore the contribution of NO to NO$_3$ loss, as indicated by equation (8). However, during the night, NO was generally below the 11 pptv detection limit and we set its mixing ratio to zero.

If we were to set the mixing ratio to 11 pptv, this would reduce the fractional contribution of VOCS to NO$_3$ detection at night. The text has been modified:

The fraction, $f$, of NO$_3$ that will react with organic trace gases is given by:

$$f = \frac{k_{OTG}^{NO_3}}{\left(\left[k_{OTG}^{NO_3}\right] + [J_{NO3}] + [NO]k_3 + K_5[NO_2]f_{het}\right)} \qquad (8)$$

where the denominator sums all loss processes for NO$_3$. Mixing ratios of NO below the 11 pptv detection limit were set to zero. Figure 9 illustrates this via a diel cycle of the median for $f$. At nighttime, $\approx$ 99% of the NO$_3$ will be lost to reaction with BVOCs, with indirect heterogeneous losses representing the remaining 1%. Note that the presence of NO at 5-10 pptv levels during the night would significantly reduce this value (see section 3.3).

Section 3.4: Should the title be "Contribution to NO3 loss"? I don't see the NOx loss discussed anywhere. We have modified the text:

At nighttime, in the absence of NO and sunlight, each NO$_3$ radical formed in the reaction of NO$_2$ with O$_3$ will either be removed indirectly via the uptake of N$_2$O$_5$ onto particles or will react with a biogenic hydrocarbon. The latter results in the formation of an organic nitrate at a yield of between 20 and 100%, depending on the specific VOC (Ng et al., 2017) and hence removal of NO$_x$.